# Estimating the Optimal Overall Slope Angle of Open-Pit Mines with Probabilistic Analysis

**Wael R. Abdellah [1,]*, Chiaki Hirohama [2], Atsushi Sainoki [3], Ahmed Rushdy Towfeek [4] and Mahrous A. M. Ali [5]**

1   Department of Mining and Metallurgical Engineering, Faculty of Engineering, University of Assiut, Assiut 71515, Egypt
2   Civil and Environmental Engineering Department, Kumamoto University, Kumamoto 860-8555, Japan; 146t4814@st.kumamoto-u.ac.jp
3   International Research Organization for Advanced Science and Technology (IROAST), Kumamoto University, Kumamoto 860-8555, Japan; atsushi_sainoki@kumamoto-u.ac.jp
4   Civil Engineering Department, Faculty of Engineering-Qena, Al-Azhar University, Qena 83513, Egypt; ahmedmahmoud.3822@azhar.edu.eg
5   Mining and Petroleum Engineering Department, Faculty of Engineering-Qena, Al-Azhar University, Qena 83513, Egypt; mahrousali@azhar.edu.eg
*   Correspondence: waelabdellah@aun.edu.eg

**Featured Application: This study employs a hybrid approach in which numerical modelling is coupled with probabilistic analysis to determine the optimal overall slope angle of an open-pit mine based on three design parameters, namely, safety, productivity and cost.**

**Abstract:** Slope instability of open-pit mines has adverse impacts on the overall mine profitability, safety and environment. The slope of an open-pit mine is crucially influenced by the slope geometry, quality of rock mass and presence of geological features and their properties. The objective of this study is to demonstrate a method to select the optimal overall slope angle of open-pit mines according to three design parameters, namely, safety (e.g., probability of instability), productivity (e.g., profit) and mining costs (e.g., cost of removal of overburden). Therefore, this study attempts a hybrid approach in which numerical modelling is integrated with probabilistic analysis to evaluate the stability of an open-pit mine at various overall slope angles. Two-dimensional elasto-plastic finite-element, RS2D, has been used to develop a series of models at different ultimate slope angles employing shear strength reduction technique (SSRT). Li's point-estimate method of $n^3$ has been invoked in deterministic analysis to tackle the inherent uncertainty associated with host rock mass properties. The results reveal that the mine profitability increases and the cost of overburden removal decreases as overall slope angle becomes steeper. However, the slope stability deteriorates. Therefore, it is highly advisable to combine these three design parameters (e.g., safety, productivity, and cost) together when selecting overall slope angle of open-pit mines.

**Keywords:** slope stability; probabilistic analysis; open-pit mines; shear strength reduction technique (SSRT); critical strength reduction factor (CSRF); open-pit mine productivity

## 1. Introduction

Stability is of the utmost importance for the success of any mining operation (e.g., surface or/and underground). It is crucial to minimize the overall mining costs while maintaining safety for better productivity. Eventually the amount of waste rock (e.g., barren dirt), which has to be removed to extract the ore, should be minimized (e.g., reducing stripping ratio). In open-pit mines, slope stability is a major concern, and its failure has negative consequences on the economy and safety of personnel and equipment. Therefore, it is necessarily required to determine the factor of safety associated with the rock slope and its possible failure pattern. Slope stability is governed by geological conditions and rock

mass characterization, which are unique to each site [1]. Estimating rock slope stability is sometimes difficult due to rock mass variability/heterogeneity (e.g., existence of joint sets, faults, anisotropy, etc.) [2–5]. Thus, the rock slope of open-pit mines, quarries and other kinds of embankments (e.g., mine tailing dams, stockpiles, waste dumps and mine tailing storage facilities, etc.) should be analyzed as "geotechnical structures".

Therefore, their stability should be evaluated according to the economic (e.g., design/select a reasonable slope angle which affects the stripping ratio and overall cost), environmental (e.g., long-term stability) and safety issues to personnel and machinery [6–8]. There are several factors controlling the rock slope stability, such as rock mass properties, the geometry of the slope (e.g., height and ultimate slope angle), the presence of groundwater and discontinuities (e.g., joints, faults, fissures). Geological structures determine the slope failure mechanism associated with rock mass and its pattern (e.g., planar, wedge, toppling and/or circular failure) [4,9–11]. Sometimes the dimensions of open pits may extend to hundreds of meters (e.g., South Africa Sandsloot, located at an elevation of 1100 m, is the largest open-pit mining platform in the world, about 1500 m long, 800 m wide and projected depth of 325 m). Consequently, millions of dollars may be lost if an inappropriate design of the overall slope angle is achieved.

Slope angle affects, more or less, the stripping ratio (e.g., the ratio of tonnage or volume of overburden to be removed to tonnage or volume of ore to be extracted) and consequently the mining profitability [9,11]. Alternatively, the higher the stripping ratio, the more expensive the mining and the less profit is then gained. The reduction of stripping ratio (e.g., less waste rock to be removed) requires maintaining the overall slope angle as steep as possible (e.g., increase the ore recovery). To do so, successful pit planning is essential and requires accurate information about geology and site characteristics (e.g., slope geometry, rock properties, groundwater conditions and associated discontinuities). The final design of the optimal slope angle is controlled not only by ore grade distribution and operational cost but also by overall rock mass properties. Therefore, it is recommended that the potential for failure be incorporated into the ultimate open pit design. It helps, in advance, to know temporally (i.e., when) and spatially (i.e., where) actions have to be taken [12]. Characteristics of rock mass are crucial design input parameters. These parameters are never known precisely. There are always uncertainties associated with them. Some of these uncertainties exist due to limited data, errors in testing, random data collection and lack of knowledge. Herein, probabilistic methods, as introduced in the next section, are used to tackle the inherent uncertainty associated with rock mass properties.

Probabilistic methods are used to deal with inherent heterogeneity/variability associated with rock mass properties. They are applied to estimate the likelihood of failure at various slope angles and take measures to reduce the risk to an acceptable level. These methods incorporate the statistical variation of the numerical model input parameters representing the rock mass properties (e.g., mean, variance and standard deviation), as well as the design of rock failure criteria [13]. In this study, Li's point-estimate method of $n^3$ (e.g., where "n" determines the number of simulations based on the model input parameters) has been invoked to study the probability of slope instability at different overall slope angles. In this investigation, our focus is the uncertainty that arises from rock mass properties (e.g., host rock mass) and their effect on the slope stability of open-pit mines. The next section discusses how to evaluate slope stability.

The objective of this study is to develop a hybrid approach in which deterministic numerical modelling is integrated with probabilistic methods to evaluate the stability of open-pit mines at various ultimate slope angles using shear strength reduction technique (SSRT).

## 2. Methods of Slope Stability Analysis

Many methods exist to assess the stability of rock slopes, and each has its merits and shortcomings. These methods are limit equilibrium methods (LEMs); analytical and finite-elements; finite differences and/or discrete element methods (FEMs/FDMs/DEMs).

LEMs (e.g., method of slices) are popular methods of analysis, where the factor of safety is obtained by dividing soil/rock mass above the hypothetical surface of failure into several vertical slices, as introduced in Equation (1). LEMs assume the location, shape of failure surface and horizontal forces acting on the sides of the slides and their directions. Despite their inherent weakness, these methods were developed and tested based on actual case histories [14–18].

$$\mathbf{FOS} = \frac{\boldsymbol{\tau_f}}{\boldsymbol{\tau_m}} \tag{1}$$

where:

$\boldsymbol{\tau_f}$ is the actual shear strength of rock mass,

$\boldsymbol{\tau_m}$ is the mean shear stress on the assumed surface of failure mobilized to maintain body in equilibrium.

According to the aforementioned Equation (1), the factor of safety (**FOS**) is defined as the ratio of soil/rock shear strength at the failure to the mobilized shear stress on the surface of failure. Due to the simplicity of LEMs, they do not consider the stress–strain behavior of the soil/rock mass when calculating the **FOS**. Also, the pattern of critical slip surface is determined by trial and error [19,20]. For more details about LEMs, the readers are referred to the work of Morgenstern and Price [21–31]. Numerical methods (e.g., FEMs, FDMs and DEMs) are powerful tools (e.g., they handle efficiently complex geometry) that provide approximate solutions to boundary value problems for partial differential equations [32–36]. They satisfy all requirements that have to be met for a complete solution to slope stability problems. The behavior of the material can be modeled with various constitutive equations and numerical simulation techniques, e.g., perfect elastoplastic analysis with Mohr–Coulomb failure criterion, creep deformation with Burger's model and jointed rock mass with interface elements [37,38]. Hybrid techniques have be- come widely acceptable in the assessment of rock slopes. These tools combined LEMs and FEMs groundwater flow and stress analysis [27,39,40]. In this study, RS2D (e.g., formerly known as Phase2D) [41–46] is used to evaluate the stability of an open pit at various general slope angles and to determine the most stable slope angle (s) employing the shear strength reduction technique (SSRT), as introduced in the next section.

### 2.1. Shear Strength Reduction Technique (SSRT)

SSRT is as an approach where the factor of safety is determined by artificially weakening or reducing soil/rock mass strength properties in steps/stages, adopting elastoplastic FEM analysis until failure/collapse of the rock slope occurs (e.g., slope fails). Numerically, failure occurs when the converged solution no longer exists. Alternatively, the factor of safety (**FOS**) is the value by which soil/rock mass has to be reduced to reach failure/collapse [47–52]. Figure 1 depicts graphically Mohr–Coulomb yield surfaces and Equation (2) presents them mathematically.

$$\mathbf{SRF} = \frac{\tan \varphi}{\tan \varphi_f} = \frac{\mathbf{C}}{\mathbf{C_f}}. \tag{2}$$

where:

**SRF** is the strength reduction factor, which is used to define the value of soil/rock mass strength parameters at a given stage of the analysis,

**C** & $\varphi$ are the soil/rock mass shear strength input values/parameters (e.g., cohesion and friction, respectively),

$\mathbf{C_f}$ & $\varphi_f$ are the soil/rock mass shear strength reduced or mobilized values used in the analysis (e.g., cohesion and friction at failure, respectively).

It is worthy to mention that SRF is set to 1.0 at the beginning of calculations (e.g., soil/rock mass strength properties are set to their input values, **C** & $\varphi$). In case of failure, the SRF, which is defined by Equation (2), corresponds to the factor of safety (**FOS**), aforementioned in Equation (1). Also, there are no assumptions required about the shape or location of the failure surface when adopting FEM methods of analysis. The failure

occurs through the zones within the soil/rock mass, in which their shear strength is unable to resist the applied shear stress. This analysis has been conducted in static drained conditions assuming effective shear strength and deformation parameters, ignoring the effect of seismicity, groundwater level and distributed load. The numerical modelling setup is presented in the following section.

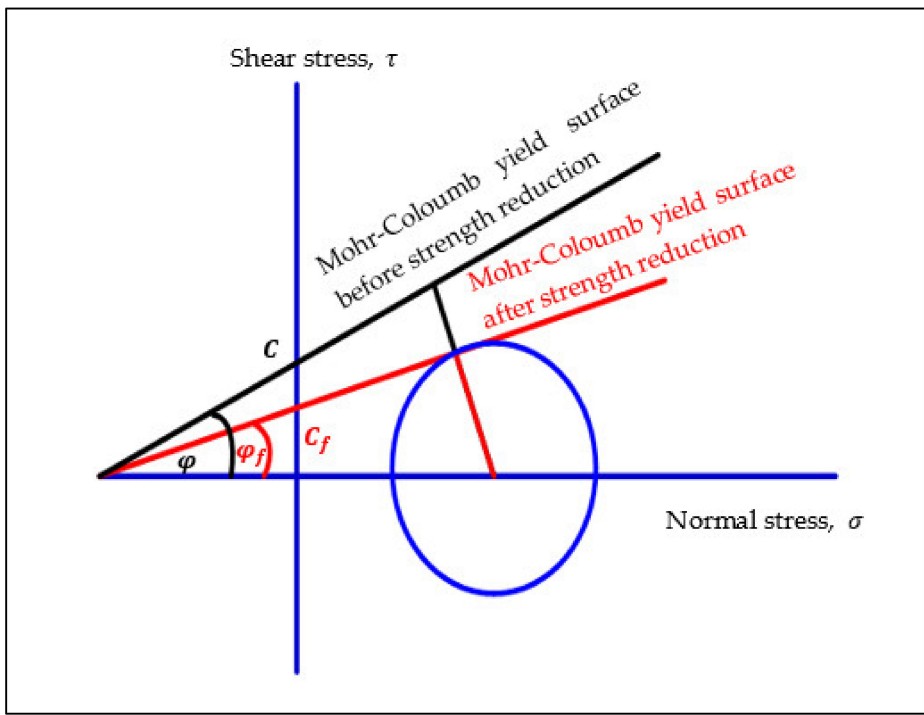

**Figure 1.** Mohr–Coulomb yield surfaces before/after strength reduction [17].

*2.2. Numerical Modelling Set Up*

Numerical modelling has been conducted using RocScience RS2D software, adopting elastoplastic Mohr–Coulomb failure criterion (M–C). The geomechanical properties for the three different rock masses, which are used in this analysis, are listed in Table 1. It is noteworthy to emphasize that this study does not represent a real case study. However, it uses representative geological properties and general open-pit mine geological features. Also, it is conducted to demonstrate a method to propose an optimal slope angle while maintaining a factor of safety (**FOS**).

**Table 1.** Mechanical properties of rock mass used in this analysis [53].

| Rock Mass | Unit Weight, kN/m$^3$ | Compressive Strength, Mpa | Elastic Modulus, GPa | Poisson's Ratio | Cohesion, Mpa | Friction Angle, (°) |
|---|---|---|---|---|---|---|
| I | 26.1 | 73.4 | 44.7 | 0.17 | 0.757 | 48.96 |
| II (Ore) | 26.1 | 40.3 | 57 | 0.20 | 0.257 | 50.15 |
| III | 27.3 | 93 | 61.2 | 0.17 | 0.245 | 48.28 |

2.2.1. Model Geometry and Boundary Conditions

To examine the slope stability of an open-pit mine at different overall slope angles, a two-dimensional model was built using RS2D software, as shown in Figure 2, followed by the other ten models at different overall slope angles. The slope height for all different models/slopes was fixed at 225 m, bench height was 15 m and the width of bench berm was 1 m. Every 45 m in height, a haul road with a width of 30 m was created. As bench height was 15 m, the open pit with a height of 225 m was excavated with 15 steps. The original

width of ore was 40 m and was represented by the green-colored part (e.g., Figure 2). However, due to model symmetry, this width was taken as 20 m. The location of the model boundaries (e.g., lateral and bottom) was determined using sensitivity analysis. The right lateral boundary was located 280 m from the top of the slope, while the bottom boundary was located 530 m below the slope toe. When boundaries are located there, they do not affect analysis results.

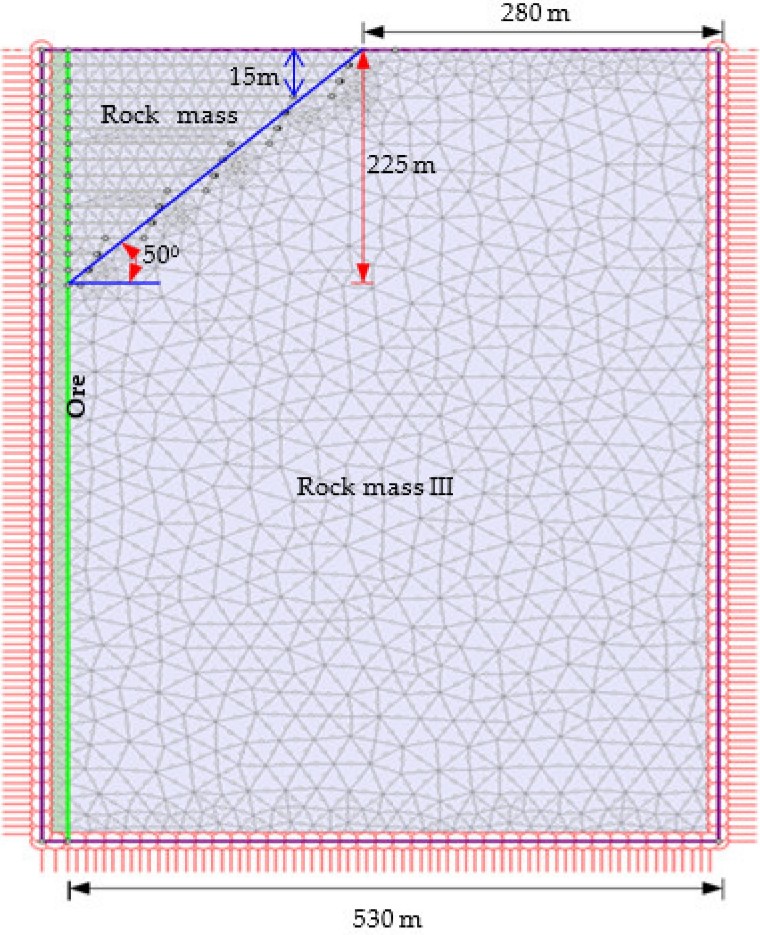

**Figure 2.** Model setup shows geometry, boundary conditions (BCs) and meshing before excavation (e.g., slope angle = 500).

### 2.2.2. Mesh Sensitivity Analysis

Conducting mesh sensitivity is highly recommended in finite elements to avoid the negative effects of very long/thin elements on analysis results. The aspect ratio of such thin elements is very high. The mesh quality option is a built-in feature in RS2D that allows users to identify and fix finite element mesh problems. Non-convergence of solution, terminated calculation, alerting or misconfigurations during computation, and extraordinary analysis results are examples of such issues [54,55]. Therefore, a sensitivity analysis was conducted to determine the optimal mesh size. Such size determines the number of nodes, elements and their suitable lengths in order to obtain consistent results. This was done by constructing models with different mesh densities. Both vertical and horizontal displacements were monitored while solving for elastic equilibrium, as shown in Figure 3.

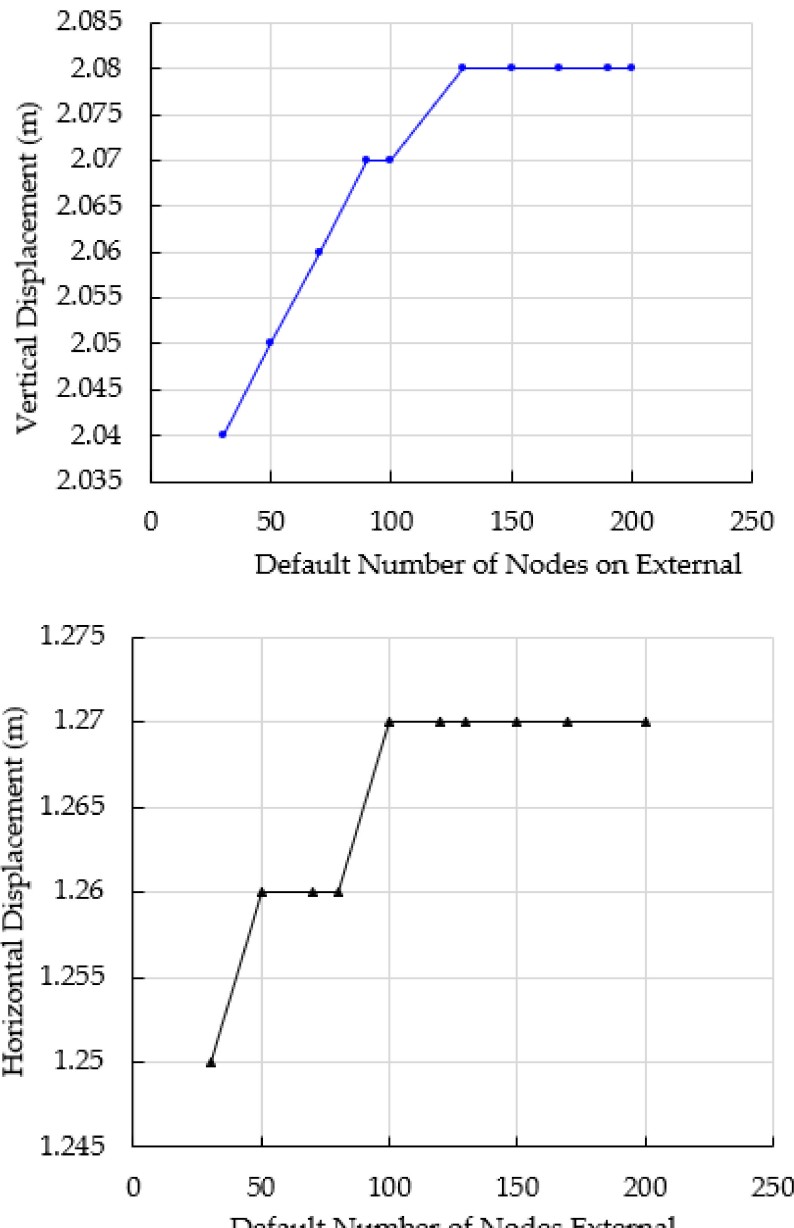

**Figure 3.** Mesh sensitivity analysis depicts the number of nodes on the external boundary versus vertical (**top**) and horizontal (**bottom**) displacements.

Several factors influence selecting mesh size or density. For instance, dense mesh provides a better representation of high-stress gradients; the accuracy of the results is proportional to element length, and if different mesh sizes are needed, then the more gradual change from dense to coarse should be employed. As shown in Figure 3, the model ran with 200 nodes when the external boundary reached a plateau.

## 3. Results

As stated before, this study applies a shear strength reduction technique (SSRT) to demonstrate a hybrid method to determine the optimal ultimate slope angle of an open pit that maintains slope stability and allows high productivity. The analysis was conducted by constructing a series of models at different overall slope angles and observing the critical strength reduction factor (SRF) or its analogy (e.g., a factor of safety, **FOS**). The results of the deterministic analysis will be presented and compared with probabilistic analysis

in terms of factor of safety (**FOS**), maximum shear strain and incremental horizontal and vertical displacements with respect to overall slope angles.

### 3.1. Deterministic vs. Probabilistic Analysis

The deterministic analysis used the average values of rock mass properties (e.g., as listed in Table 1). Such analysis has proven to be extremely useful for understanding and predicting the mechanical behavior of rock mechanics. The main concern in the application of deterministic modelling generally arises from the uncertainties affecting the mechanical properties of materials and field stresses, which must somehow be introduced in the analysis. In many instances, these parameters should be considered as random variables or random fields. Therefore, Li's point-estimate method has been adopted herein to tackle the inherent uncertainties arise from rock mass properties. This method includes varying rock mass properties (e.g., rock mass I, due to its close proximity to the orebody and slope location) based on predetermined distribution and specified coefficient of variation ($\delta$). The means ($\mu$) and standard deviations ($\sigma$) of these values were picked from a normal distribution. Twenty-seven runs were completed to analyze the performance criteria from the model outputs: the factor of safety, maximum shear strain and incremental vertical and horizontal displacements. Based on the parametric study (sensitivity analyses) that was conducted, the most influential model input parameters are Young's modulus (E), cohesion (C) and angle of internal friction ($\phi$), as listed in Table 2.

**Table 2.** Random properties for rock mass I [56].

| Rock Mass Property | Mean ($\mu$) | Standard Deviation ($\sigma$) | Coefficient of Variation ($\delta = \frac{\sigma}{\mu}$) |
|---|---|---|---|
| Cohesion (C), Mpa | 0.757 | 0.151 | 0.20 |
| Friction angle ($\phi$), deg. | 48.96 | 9.79 | 0.20 |
| Young's modulus (E), GPa | 44.70 | 8.94 | 0.20 |

3.1.1. Factor of Safety (FOS)

The factor of safety (**FOS**) or critical strength reduction factor (CSRF) refers to the value by which soil/rock mass strength properties (e.g., cohesion, **C** & friction, $\varphi$) have to be reduced to reach failure. Figure 4 shows the factor at various slope angles. The factor of safety of 1.5 was decided as the threshold that corresponds to the safe ultimate slope angle. It can be seen that the factor of safety decreases as the slope angle increases. The deterministic analysis shows that, based on the selected threshold (e.g., **FOS** = 1.5) [57], it is not safe to design an open-pit mine in which the overall slope angle exceeds 45°. Probabilistic analysis shows that it is unsafe to design an open-pit mine with an overall slope angle beyond 50°. It is noteworthy to mention that the deterministic analysis gives only a single output, which does not give any information about the variability of the input variable (e.g., no distribution). However, probabilistic analysis is considered most beneficial and accurate with limited data (e.g., there is a distribution for the input variable). The probability density function (PDF) of the factor of safety after 27 simulations (e.g., using Li's PEM), at various overall slope angles, is displayed in Figure 5.

As illustrated in Figure 5, the area under the curve that corresponds to **FOS** < 1.5 increases as the slope angle increases. The probability of failure, P*f*, at different overall slope angles, based on the estimated factor of safety (**FOS**), will be presented and discussed later (e.g., Section 3.2).

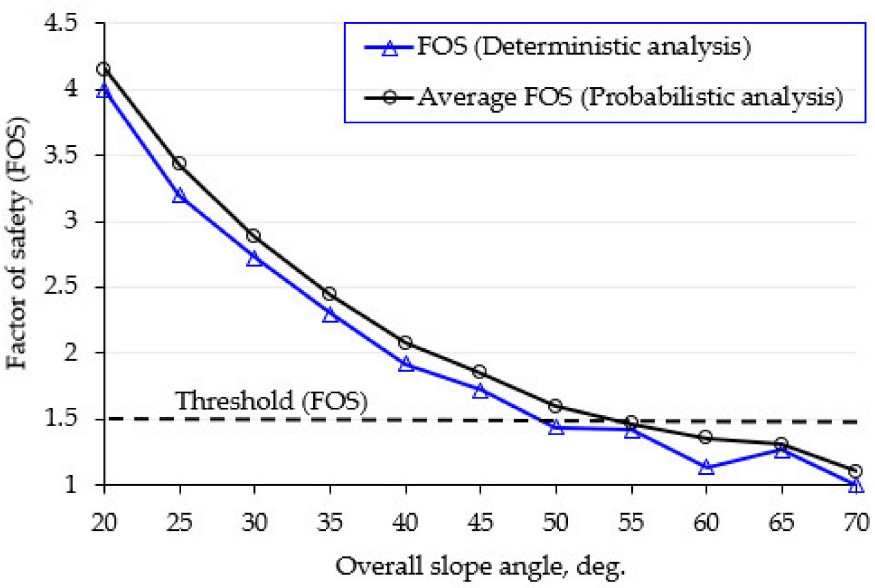

**Figure 4.** Factor of safety versus different overall slope angles (deterministic vs. probabilistic analysis).

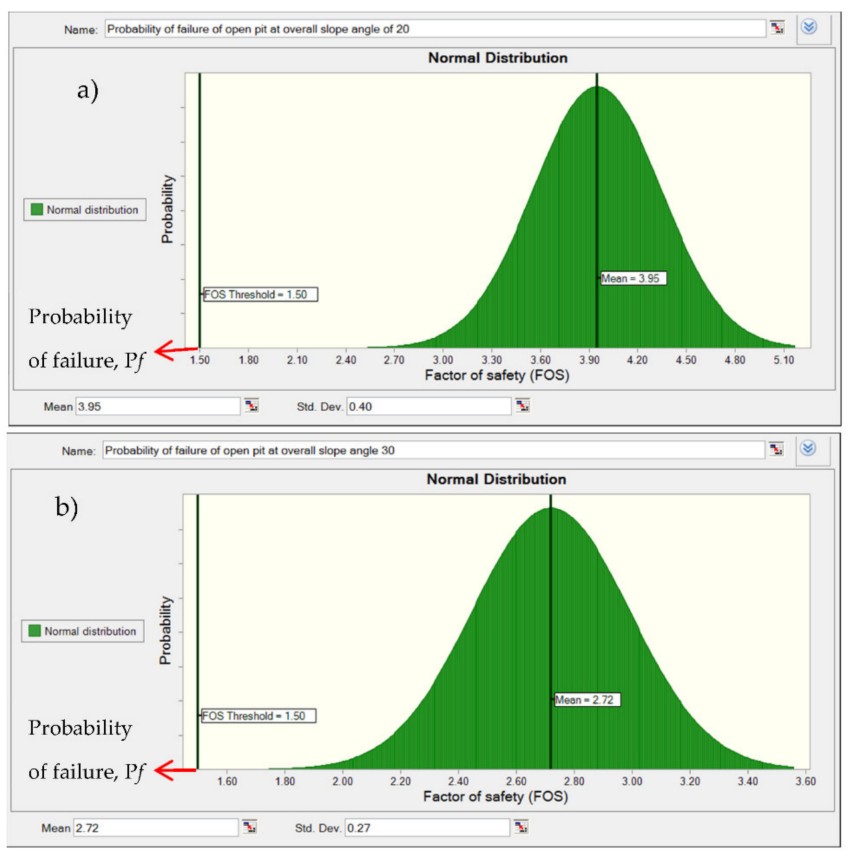

**Figure 5.** *Cont.*

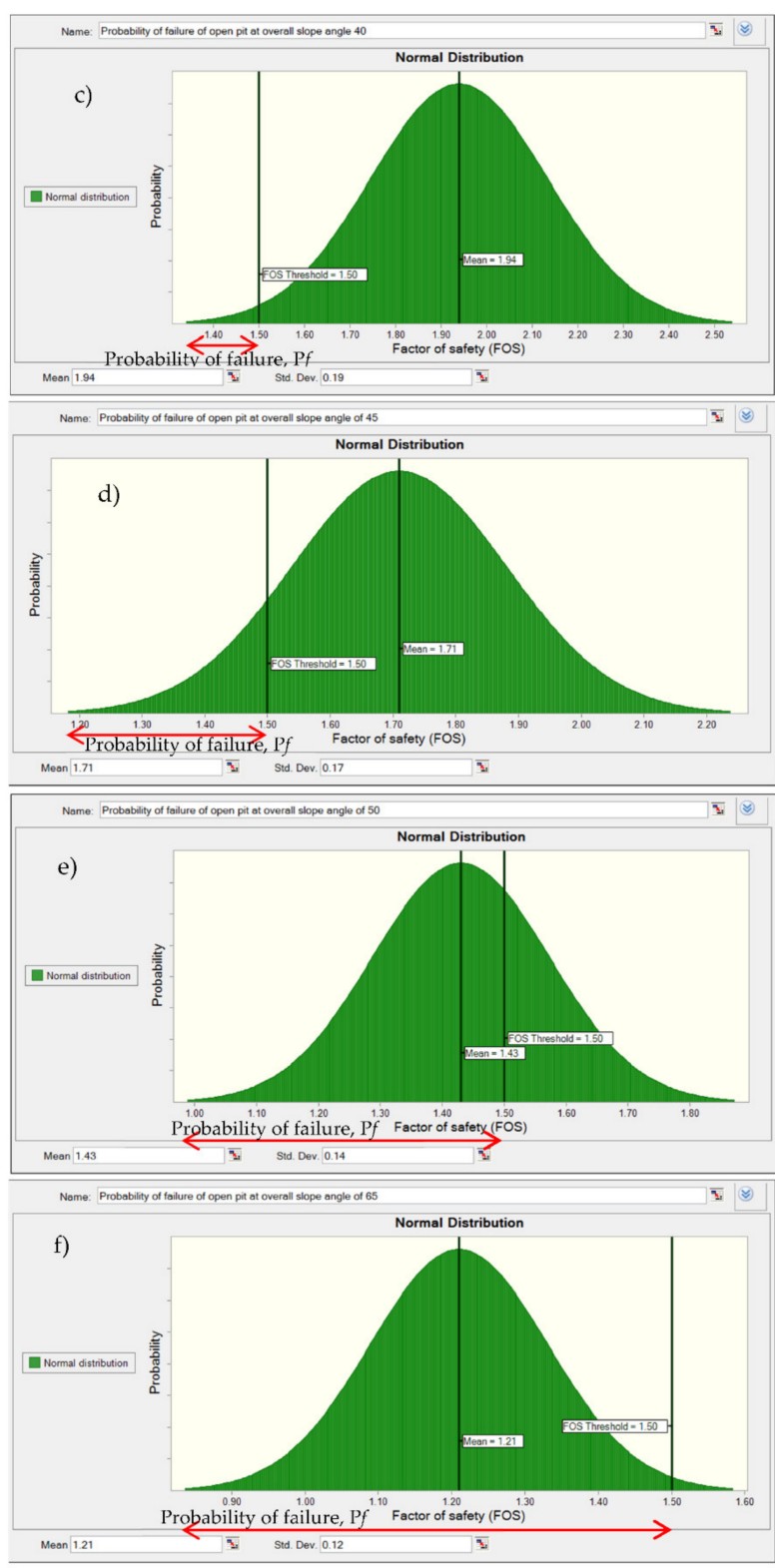

**Figure 5.** *Cont.*

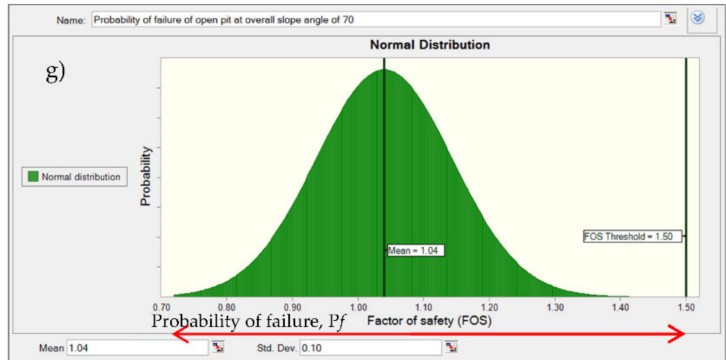

**Figure 5.** Probability density function (PDF) curves for factor of safety (**FOS**) at various overall slope angles after 27 simulations. (**a**) P$f$ at overall slope angle of 20°, (**b**) P$f$ at overall slope angle of 30°, (**c**) P$f$ at overall slope angle of 40°, (**d**) P$f$ at overall slope angle of 45°, (**e**) P$f$ at overall slope angle of 50°, (**f**) P$f$ at overall slope angle of 65° and (**g**) P$f$ at overall slope angle of 70°.

### 3.1.2. Maximum Shear Strain

Figure 6 depicts the maximum shear strain at different slope angles. It can be seen that the shear strain increases as the slope angle increases. Also, the probabilistic analysis gives higher values of shear strain compared to deterministic analysis. Figure 7 presents the probability density function (PDF) curves for the obtained shear strain at various overall slope angles.

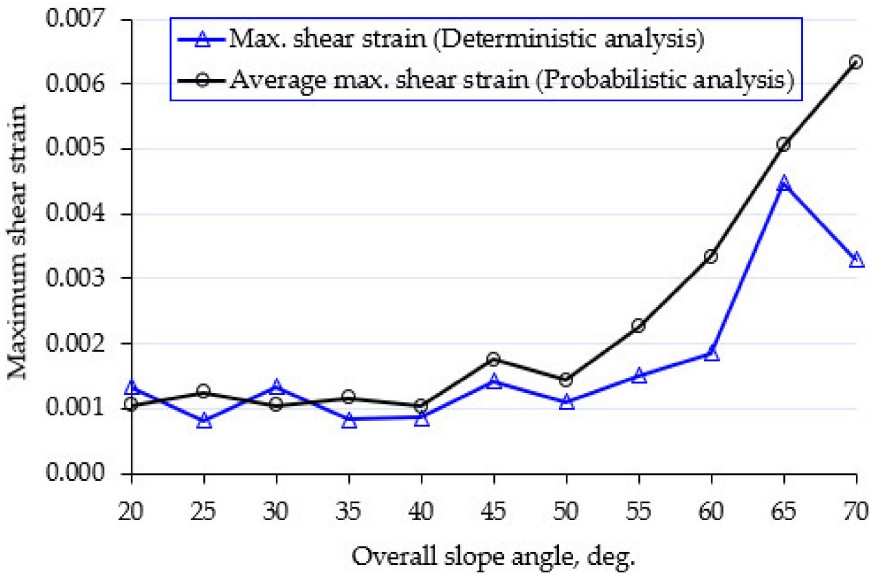

**Figure 6.** Maximum shear strain at different overall slope angles (deterministic vs. probabilistic analysis).

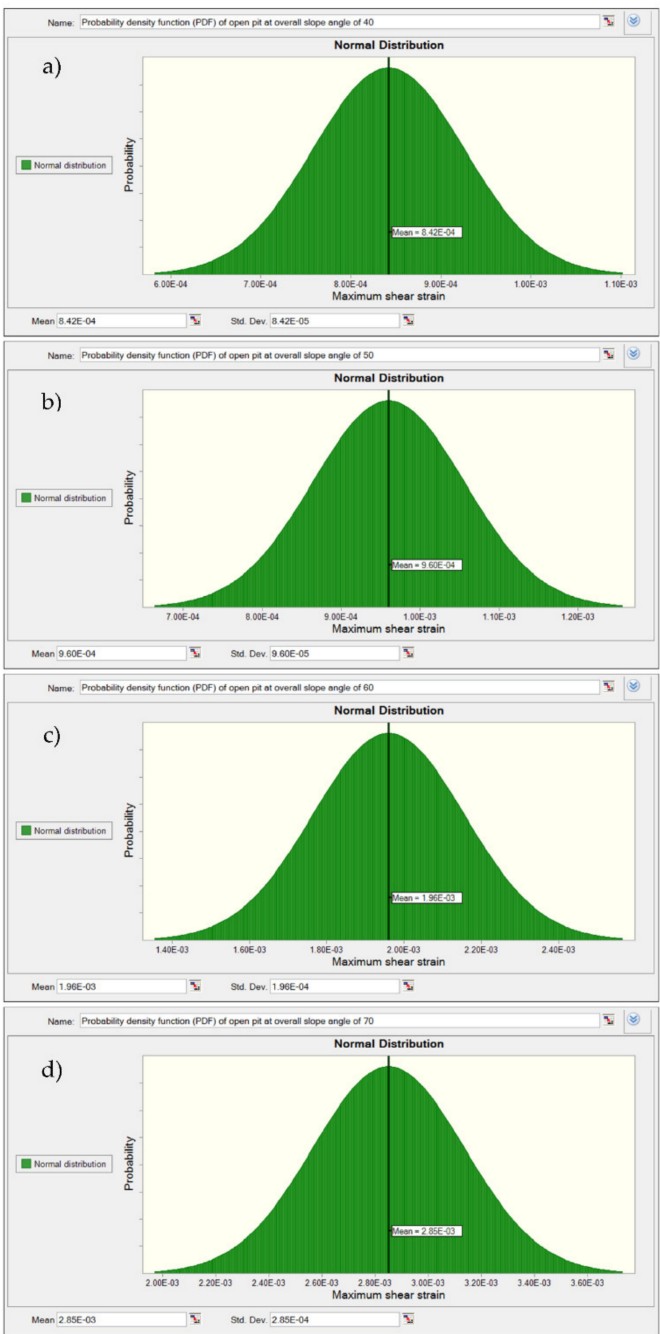

**Figure 7.** Probability density function (PDF) curves for maximum shear strain at different overall slope angles: (**a**) P$f$ at overall slope angle of 40°, (**b**) P$f$ at overall slope angle of 50°, (**c**) P$f$ at overall slope angle of 60° and (**d**) P$f$ at overall slope angle of 70°.

### 3.1.3. Incremental Horizontal and Vertical Displacements

Figure 8 presents the incremental horizontal and vertical displacements that have been recorded at each mining stage (e.g., in total 15 mining stages). It is obvious that the increase in the slope angle leads to an increase in both incremental horizontal and vertical displacements. The probabilistic analysis gives high magnitudes of both incremental displacements compared with deterministic analysis. Figures 9 and 10 present the probability density function (PDF) curves of incremental horizontal and vertical displacements at various slope angles, respectively.

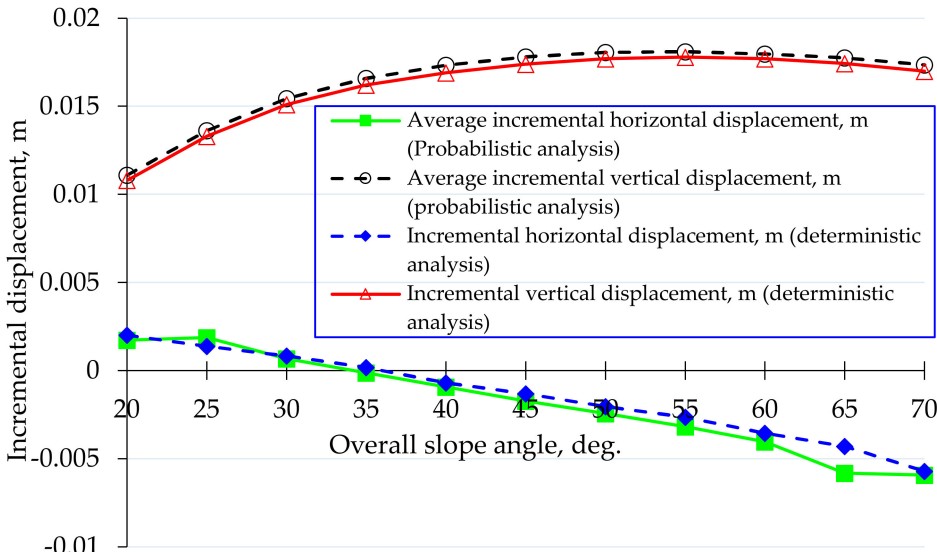

**Figure 8.** Incremental displacements at various overall slope angles (deterministic vs. probabilistic analysis).

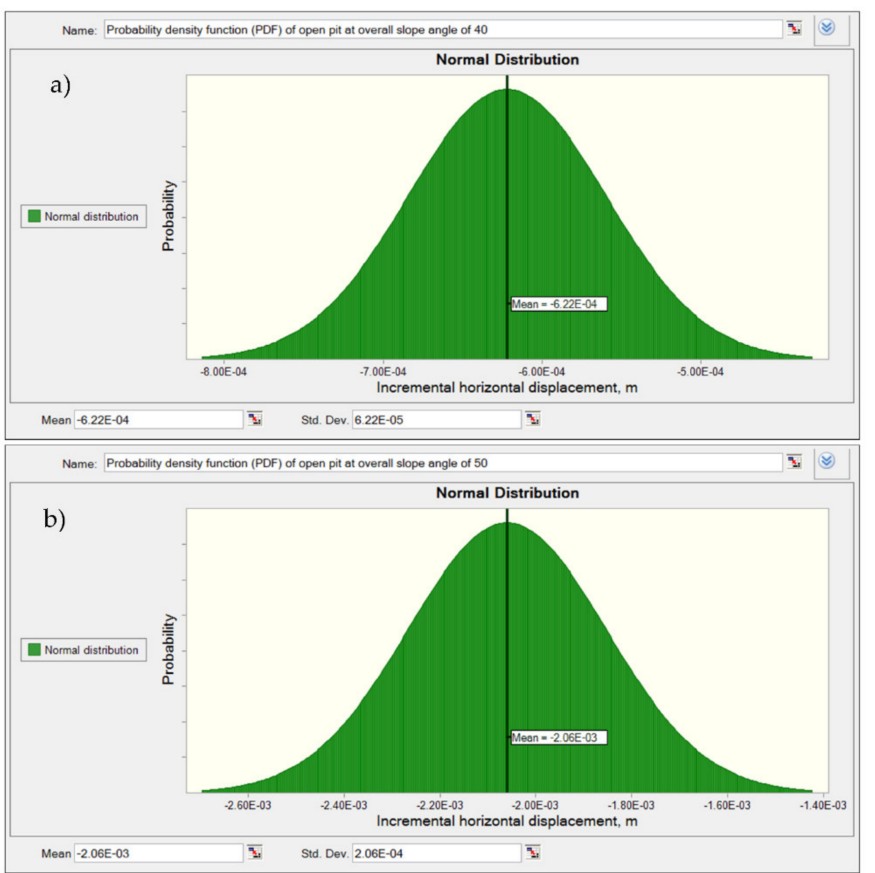

**Figure 9.** *Cont.*

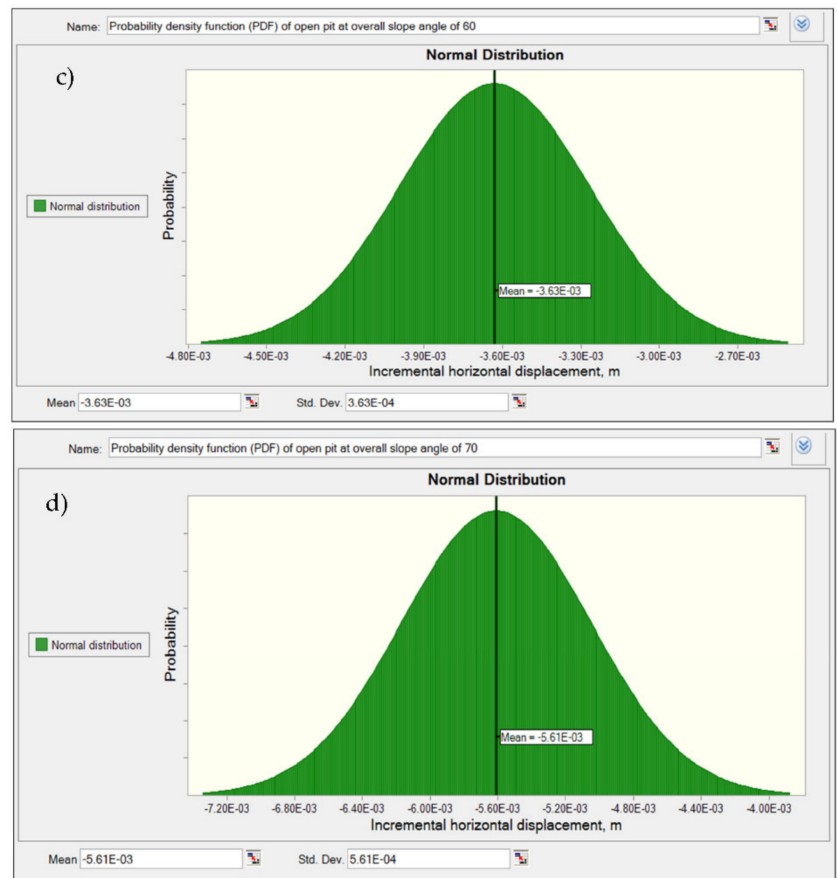

**Figure 9.** Probability density function (PDF) of incremental horizontal displacement at different overall slope angles: (**a**) P*f* at overall slope angle of 40°, (**b**) P*f* at overall slope angle of 50°, (**c**) P*f* at overall slope angle of 60° and (**d**) P*f* at overall slope angle of 70°.

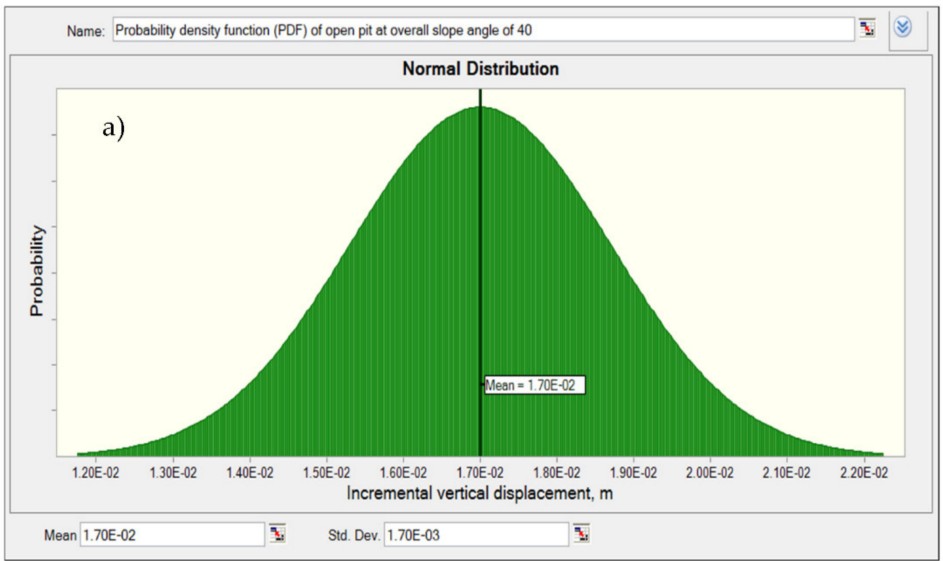

**Figure 10.** *Cont.*

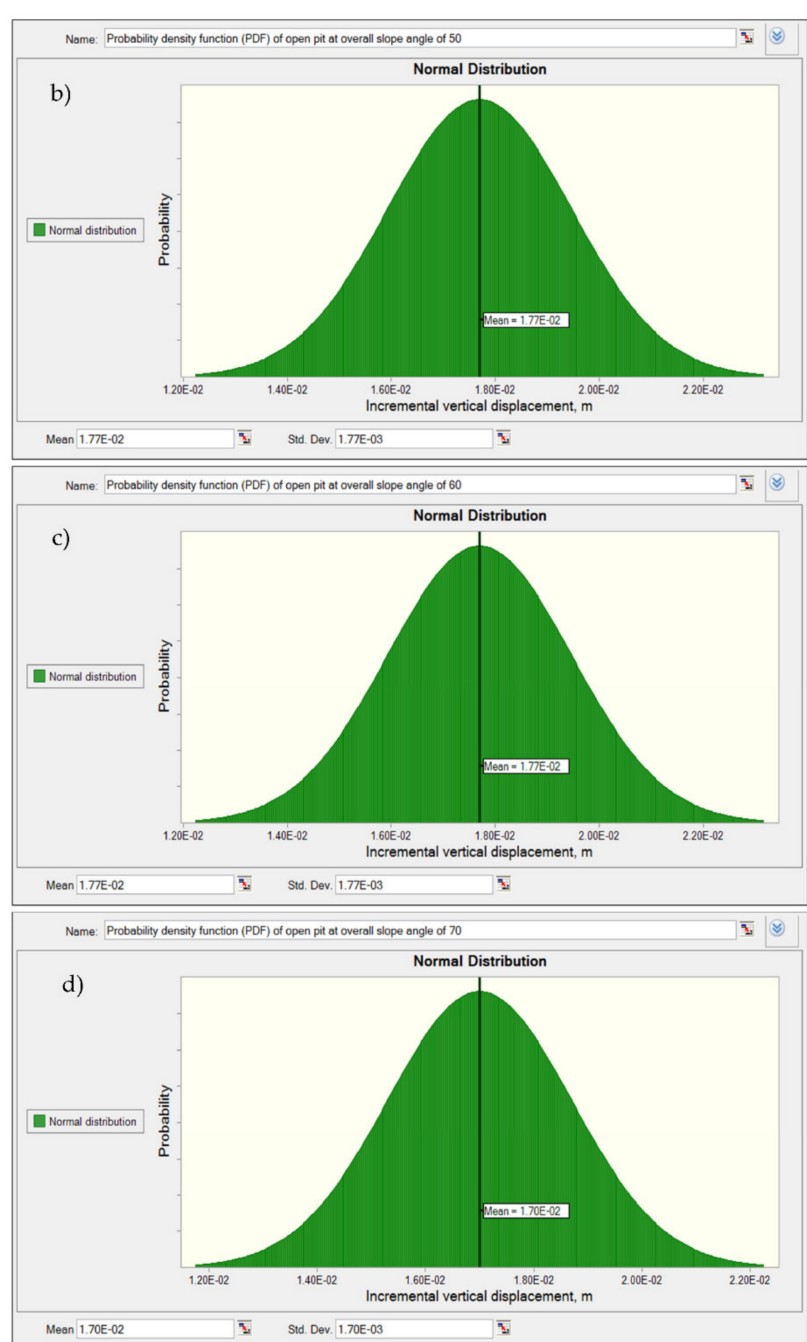

**Figure 10.** Probability density function (PDF) of incremental vertical displacement at different overall slope angles: (**a**) P*f* at overall slope angle of 40°, (**b**) P*f* at overall slope angle of 50°, (**c**) P*f* at overall slope angle of 60° and (**d**) P*f* at overall slope angle of 70°.

### 3.2. Estimating the Probability of Failure

The probability of instability, P*f*, is estimated using Z-tables (e.g., standard normal variate). Such tables, Z-tables [56,58], are used to determine the area under probability density function (PDF) curves. The suggested ratings of likelihood and ranking of the probability of instability, P*f*, are listed in Table 3. Figure 11 depicts the likelihood of failure at various slope angles. It can be seen that the probability of instability, P*f*, increases as the overall slope angle increases.

**Table 3.** Suggested ratings of likelihood and ranking of probability of failure, P*f* [56].

| Rating | Likelihood Ranking | Probability of Occurrence | |
|---|---|---|---|
| 1 | Rare | <5% | May occur in exceptional circumstances. |
| 2 | Unlikely | 5–20% | Could occur sometimes |
| 3 | Possible | 20–60% | Might occur sometimes |
| 4 | Likely | 60–85% | Will probably occur in most circumstances |
| 5 | Certain | >85% | Expected to occur in most circumstances |

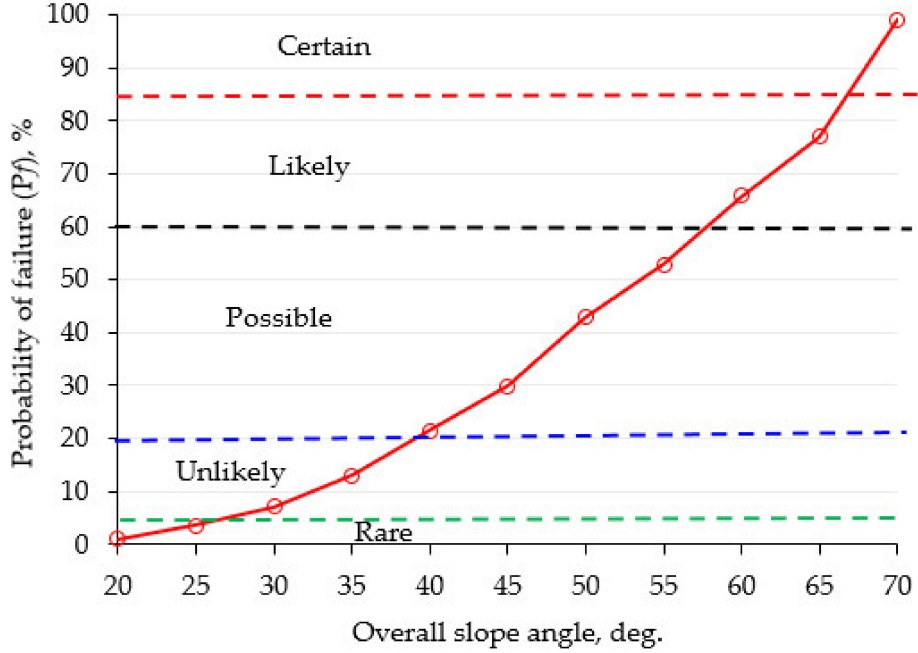

**Figure 11.** Probability of failure, P*f*, at various slope angles based on factor of safety (e.g., at **FOS** threshold = 1.5).

### 3.3. Productivity vs. Mining Costs

The amount of ore to be extracted and the cost of removal of overburden which have been estimated at different slope angles are listed in Table 4. The results reveal that the mine profitability increases and the cost of overburden removal decreases as the overall slope angle becomes steeper. However, the slope stability deteriorates, as shown in Figure 12.

**Table 4.** The value of mineral extracted and the corresponding overburden to be removed.

| Overall Slope Angle, Deg. | Ore | | | Overburden | | | Net Profit, $/ton (Price Ore-Price Overburden) $\times 10^4$ | P*f*, % |
|---|---|---|---|---|---|---|---|---|
| | Vol., m³ | Ton ($\gamma$ = 2.61 t/m³) | Price, ($100/ton) $\times 10^4$ | Vol., m³ | Ton ($\gamma$ = 2.61 t/m³) | Price, (5$/ton) $\times 10^4$ | | |
| 20 | | | | 69,545.52 | 181,513.81 | 90.76 | 144.14 | 0.89 |
| 25 | | | | 54,282.83 | 141,678.19 | 70.84 | 164.06 | 3.44 |
| 30 | | | | 43,842.54 | 114,429.03 | 57.21 | 177.69 | 7.08 |
| 35 | | | | 36,150.00 | 94,351.50 | 47.18 | 187.72 | 12.92 |
| 40 | | | | 30,166.26 | 78,733.94 | 39.37 | 195.53 | 21.48 |
| 45 | 9000 | 23,490 | 234.9 | 25,312.50 | 66,065.63 | 33.03 | 201.87 | 29.81 |
| 50 | | | | 21,239.71 | 55,435.64 | 27.72 | 207.18 | 42.86 |
| 55 | | | | 17,724.00 | 46,259.64 | 23.13 | 211.77 | 52.79 |
| 60 | | | | 14,614.18 | 38,143.01 | 19.07 | 215.83 | 65.91 |
| 65 | | | | 11,803.41 | 30,806.90 | 15.40 | 219.50 | 77.04 |
| 70 | | | | 9213.00 | 24,045.92 | 12.02 | 222.88 | 99.04 |

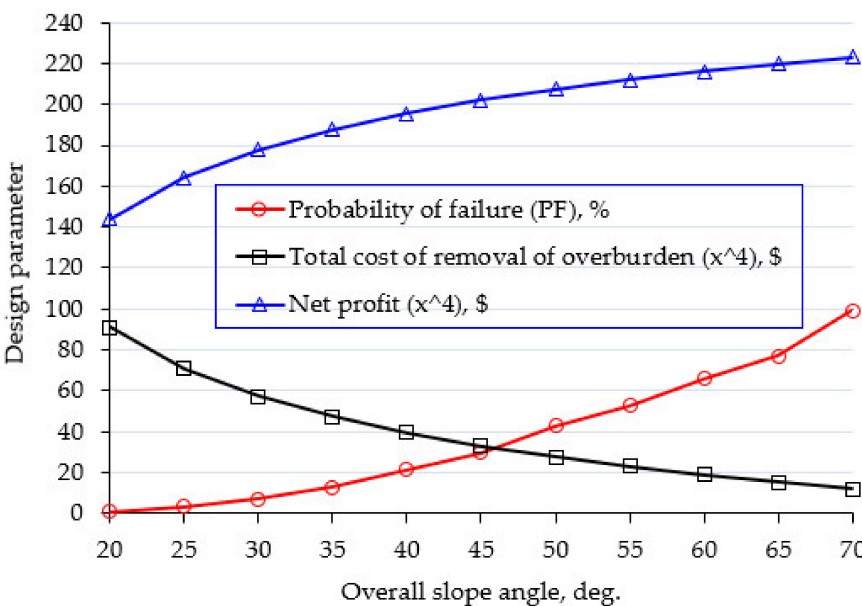

**Figure 12.** The combined three design parameters at different overall slope angles.

## 4. Discussion

The likelihood of failure, $Pf$, of an open pit increases as the slope angle increases. Geotechnical, economic, technical, strategic and regulatory compliance are all important factors in slope optimization [59–62]. Generally, the slope stability of an open pit deteriorates as shear strain increases. However, it is difficult to set a threshold for maximum shear strain. Thus, it cannot be used as a failure evaluation criterion for an open pit (e.g., probability of failure, $Pf$, cannot be estimated). For incremental horizontal and vertical displacements, the findings revealed that the rock mass moves towards ore direction (e.g., negative lateral displacement due to mining activity) as the slope angle increases. The probability of failure due to rock mass movement, on the other hand, will not be estimated because establishing a threshold is difficult. The use of rock mass deformation would be beneficial if multi-point hole extensometers (MPBHXs) were installed to monitor and read the rock mass deformation. As a result, it can be used to validate the readings with numerical analysis. Alternatively, incremental displacement won't be employed as a stability indicator (e.g., no threshold has been decided). The Inverse Velocity (IV) method has been successfully employed in conjunction with slope-monitoring radars to predict slope failure. However, the application of this method is limited due to inherent uncertainty associated with rock mass [60].

The probability of failure is rare (e.g., $Pf < 5\%$) when overall slope angles are small (e.g., 20° and 25°). Then, the probability of instability becomes unlikely when overall slope angles are 30° and 35° (e.g., $5\% < Pf < 20\%$). When slope angles lie between 40° and 55°, the probability of failure becomes possible (e.g., $20\% < Pf < 60\%$). At steep overall slope angles (e.g., 60° and 65°), the probability of failure is likely (e.g., $60\% < Pf < 85\%$). However, the probability of failure becomes certain (e.g., $Pf > 85\%$ at very steep angles (e.g., 70°) [63]. Therefore, it is recommended not to design an open pit at steep angles (e.g., $\geq 60°$), as the probability of failure is likely to certain. When the slope angle is 60°, the mine net profitability and cost of overburden removal are $ 2,158,300 and $190,70°, respectively. However, the probability of instability is likely (e.g., $Pf = 65.91\%$), whereas at a slope angle of 20°, the mine net profitability cost of overburden removal and the probability of instability are $1,441,400, $ 907,600 and 0.89% (e.g., rare). Therefore, it is necessary to combine these three design parameters together when selecting/designing the overall slope angle of the open pit [64].

*Case Study*

The open pit case study was situated in the centre of the Andes mountains of Peru at an elevation of 5000 m. Limestone is the primary lithology, whereas the ore deposit is a polymetallic (e.g., zn, pb and Ag). Skarn rock comes into contact with an ore deposit at the bottom. The host rock (limestone) has a length of 172 m, width of 202 m and projects 50 m downward. The dimensions of orebody are 170 m × 100 m (L × W), and it extends 75 m to the bottom. The skarn is located at the base of the ore deposit. Figure 13a,b show a three-dimensional perspective view and a two-dimensional sectional plan. The geomechanical properties of the open pit rock masses are listed in Table 5.

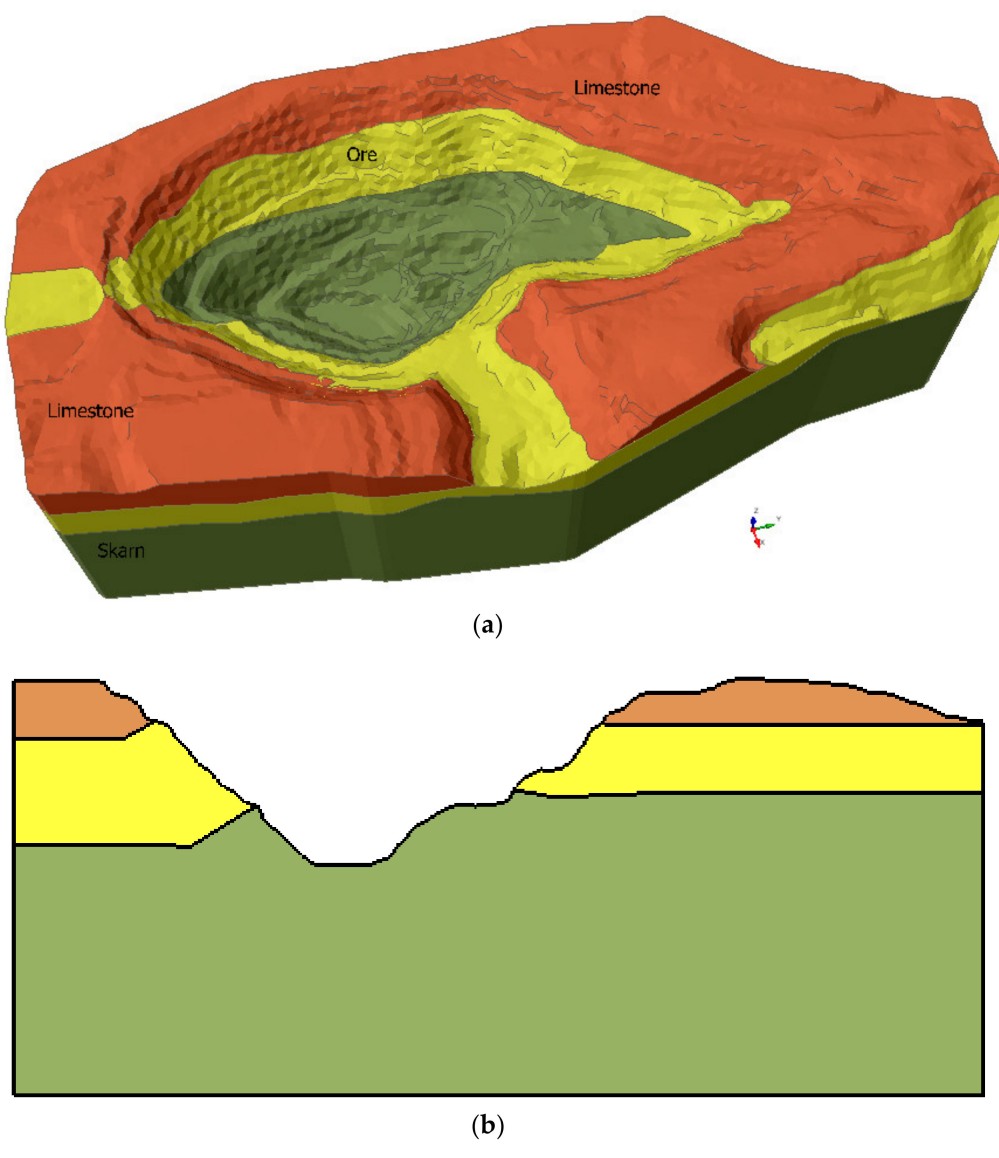

(a)

(b)

**Figure 13.** (**a**) 3D perspective view of open pit case study. (**b**) 2D plan section.

**Table 5.** Geomechanical properties of rock masses of open pit case study [57].

| Rock Mass | Unit Weight, kN/m$^3$ | Compressive Strength, Mpa | Elastic Modulus, GPa | Poisson's Ratio | Cohesion, Mpa | Friction Angle, Deg. |
|---|---|---|---|---|---|---|
| Limestone | 27 | 80 | 55 | 0.25 | 4.8 | 40 |
| Ore | 25 | 50 | 57 | 0.20 | 2.5 | 25 |
| Skarn | 30 | 85 | 65 | 0.18 | 4.6 | 35 |

Figure 14 depicts the contours of the factor of safety (**FOS**) for the open pit case study. There are three failure surfaces visible; two of them failed (e.g., **FOS** < 1.5). The minimum **FOS** of deterministic analysis is 1.414. The groundwater effect was taken into account during the analysis. For example, the pore fluid unit weight is 9.81 kN/m$^3$ and the rapid drawdown method is Duncan, Wright, Wong (1990) [65]. Figure 15 depicts the contours of slip surfaces after accounting for the groundwater effect. Three slip surfaces were noticed, with a minimum **FOS** of 1.265.

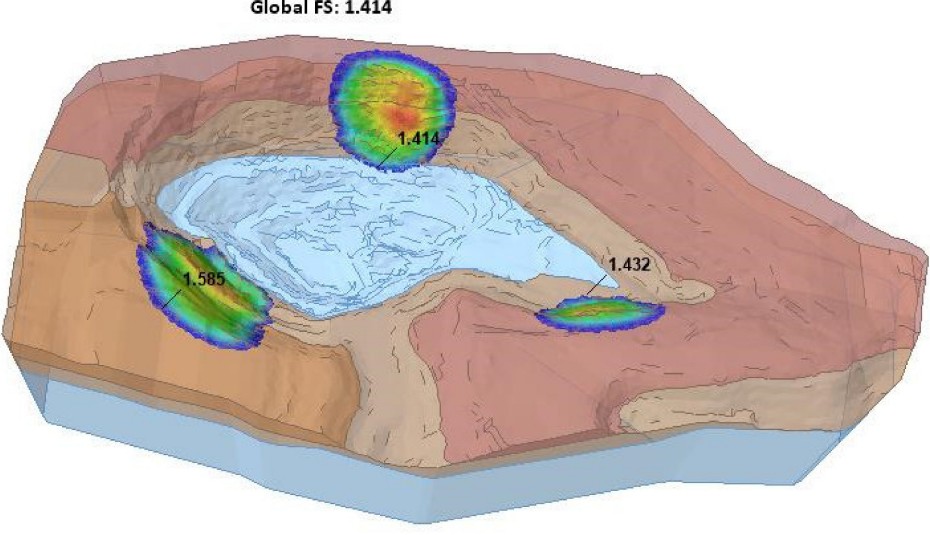

**Figure 14.** Contours of multi-material slip surfaces (deterministic analysis).

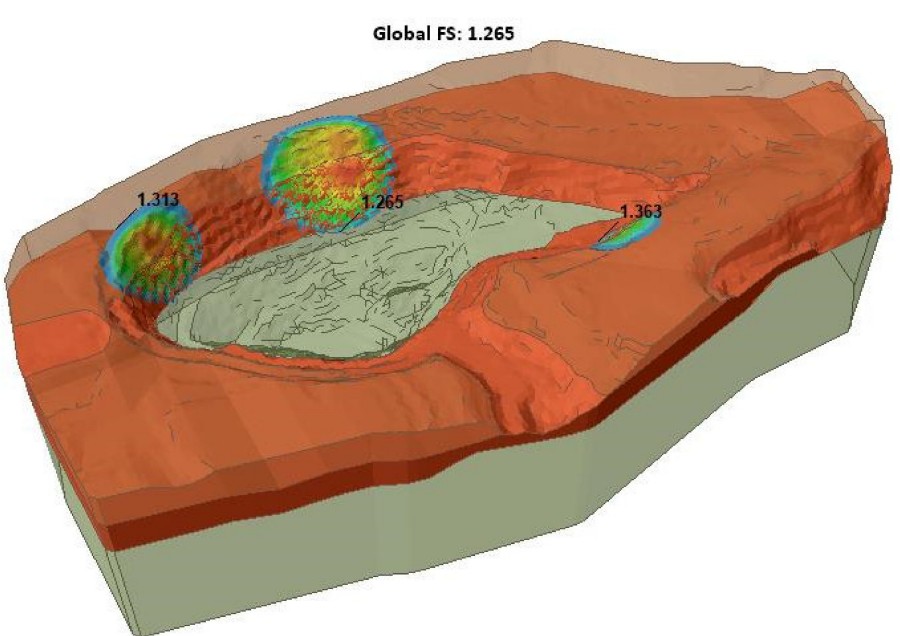

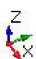

**Figure 15.** The contours of multi-material slip surfaces due to the effect of groundwater.

The effect of a pseudostatic seismic load on the minimum safety factor was also investigated in this analysis. The critical seismic coefficient (ky) that results in an unstable slope with **FOS** = 1.5 has been calculated for all failure surfaces. The critical slip surface is depicted in Figure 16 along with the critical seismic coefficient (ky = 0.154).

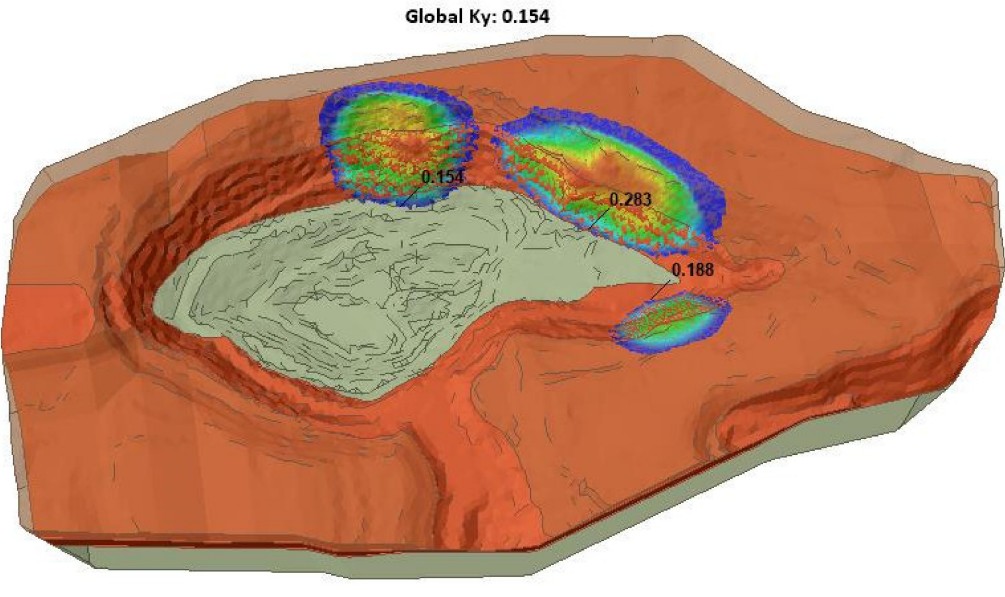

**Figure 16.** The critical seismic coefficient contours for multi-material slip surfaces.

The probabilistic analysis was carried out in order to address the inherent uncertainty associated with rock mass. Figure 17 shows one slip surface with an **FOS** of 1.246 and a failure probability of 16%. Figure 18 depicts the factor of safety (**FOS**) distribution for Monte Carlo simulations. Based on this, the probability of pit slope failure P[**FOS** < 1.5] is 16%. According to Table 3 [56], the calculated Probability of Pit Slope Failure (PoF) is unlikely, but it could happen at any time.

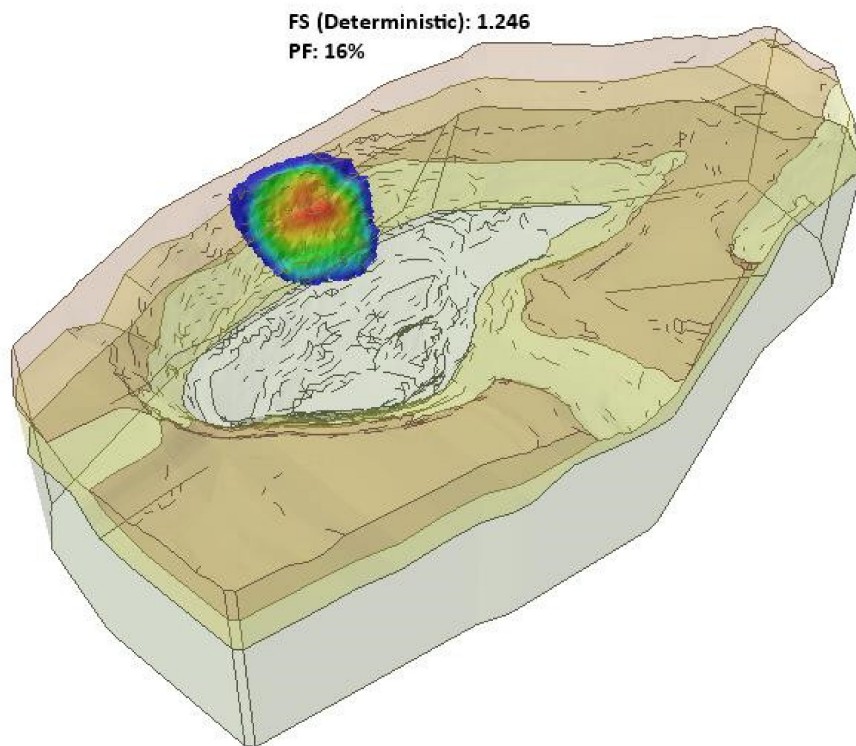

**Figure 17.** Contours of failure surface (probabilistic analysis).

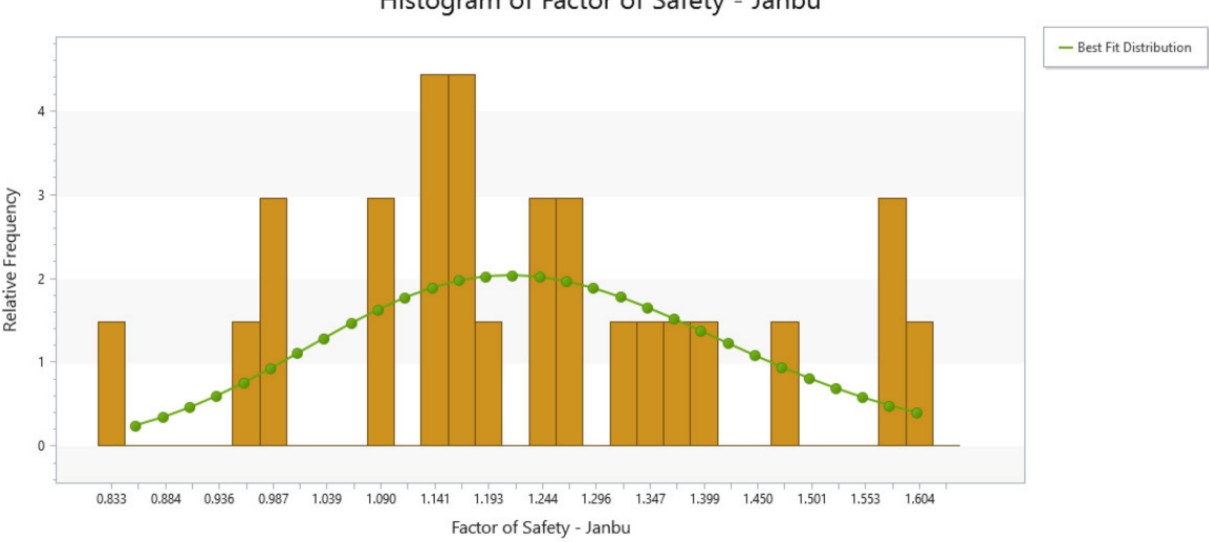

**Figure 18.** Histogram of the FoS from the 3D probabilistic analysis.

The overburden volume is $1.74 \times 10^6$ m$^3$ ($4.7 \times 10^6$ ton), while the ore deposit volume is $1.28 \times 10^6$ m$^3$ ($3.2 \times 10^6$ ton). Assume that the cost of removing one ton of overburden and recovering one ton of ore is \$5/ton and \$100/ton, respectively. As a result, the total costs of overburden removal and ore recovery are $\$23.5 \times 10^6$ and $\$320 \times 10^6$, respectively. It is important to note that the total operating cost includes both the cost of ore extraction (e.g., $\$320 \times 10^6$) and the cost of overburden removal (e.g., $\$23.5 \times 10^6$). As a result, the total operating cost is $343.5 \times 10^6$. If the selling price of one ton of ore is \$200, then the total selling price will be $\$6.4 \times 10^9$. Consequently, the net profit equals total selling price minus total operating costs (e.g., $\$64 \times 10^9 - \$343.5 \times 10^6 = \$63.6565 \times 10^9$).

## 5. Conclusions

This paper proposes a methodology for evaluating open-pit mine stability at various overall slope angles in terms of safety, productivity, and cost. Due to the heterogeneity of the rock mass, a hybrid approach method was used, in which deterministic numerical modelling was combined with probabilistic methods. At various slope angles, three failure evaluation factors were investigated: factor of safety (**FOS**), maximum shear strain and incremental displacements. The slope failure probability was calculated using a minimum FoS of 1.5. When overall slope angles range from 20° to 35°, the results show that the probability of slope failure, P$f$, is rare to unlikely (e.g., 5% < P$f$ < 20%). As the slope angle increases (e.g., from 40° to 55°), the probability of slope failure, P$f$, increases (e.g., 20% < P$f$ < 60%). At steep slope angles, the probability of slope failure, P$f$, increases (e.g., 60% < P$f$ < 85%) (e.g., 60° and 65°). The probability of slope failure, P$f$, is certain at very steep slope angles (e.g., 70°) (e.g., P$f$ > 85%). For safety purposes, designing open pits at steep to very steep angles (e.g., ≥60°) is not recommended. In terms of productivity and profitability, the results show that mine productivity increases and the cost of overburden removal decreases as the slope angle increases. However, slope stability suffers. For instance, at the steepest slope angle (e.g., 70°) the mine net profitability, cost of overburden removal and probability of instability, P$f$, are \$2,228,800, \$120,200 and 99.04%, compared to \$1,441,400, \$907,600 and 0.89% at a slope angle of 20°, respectively.

The proposed method was replicated using a real case study open-pit mine. The effect of groundwater and seismicity was examined. The results show that there are multiple unstable slip surfaces. Because of the influence of groundwater, the minimum **FOS** is 1.265. Seismicity demonstrates that **FOS** decreases to 0.154. (e.g., critical seismic coefficient). Deterministic analysis reveals multiple slip surfaces, two of which fail. Probabilistic

analysis, on the other hand, produces only one failed slip surface. Uncertainty associated with rock mass reduces **FOS** from 1.414 (deterministic) to 1.246 (probabilistic), with a failure probability of 16%. In terms of cost and profitability, the total estimate of overburden removal and ore recovery are $23.5 \times 10^6$ and $320 \times 10^6$, respectively. Consequently, the net profit will be $296.5 \times 10^6$.

**Author Contributions:** Conceptualization, W.R.A., C.H. and M.A.M.A.; methodology, W.R.A., C.H. and M.A.M.A.; investigation, W.R.A., A.S., A.R.T. and M.A.M.A.; writing—original draft preparation, W.R.A., A.R.T., A.S. and M.A.M.A.; writing—review and editing, W.R.A., A.R.T. and C.H.; visualization, W.R.A., M.A.M.A., A.S. and A.R.T.; supervision, W.R.A., A.R.T., C.H. and M.A.M.A. All authors have read and agreed to the published version of the manuscript.

**Funding:** This research received no external funding.

**Institutional Review Board Statement:** Not applicable.

**Informed Consent Statement:** Not applicable.

**Data Availability Statement:** Not applicable.

**Conflicts of Interest:** The authors declare no conflict of interest.

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
