# Peer review of "Estimating the Optimal Overall Slope Angle of Open-Pit Mines with Probabilistic Analysis"

_applsci, doi:10.3390/app12094746_

Round 1

Reviewer 1 Report

Line 176 - give reference to your statement about sensitivity analysis! 

Line 183 - Section 2.2.2; how do you perform the sensitivity analysis, we want to know some references! I think it is inbuilt with the RS2D model, if so please mentioned it! and also discussed what the sensitivity analysis tells you!

Section 3, Results:

Line 203- earlier you mentioned that you are performing probabilistic analysis but how does the deterministic come here? Please correct it here or earlier or both with the new statement! Ok, you did both - hybrid!!! Li’s point-estimate is a probability approach.

Section 3.1.1, Line 227-228- do you have any reason or have any reference for taking 1.5 FOS as threshold slope? provide ref. 

Line 232 - check "slope angle beyond 500" is correct!

Figure 5, Line 345-Give sub-script for each figure and describe what the fig. is telling.

Figure 7, Line 262 - same as above in Fig. 5.

Figure 9, 10 - same as above Figs.

Section 3.2, Line need ref. for the Z-table;

Section 3.3, I think you need to mention the value of your mineral extracted and show the differences in the table with corresponding overburden or spoils to be removed! The graph is fine but this is not able to explain what you are saying!

Section 4, Discussion - I don't see any references in this section; go ahead and find some work relevant to this work and made a comprehensive discussion and how others are dealing with this type of issues!

Line 304, this line is repeated earlier in the results section, remove it or needed to restructure;

Line 310, Give the reason why the probability of failure due to rock mass movement was not estimated? I think this is important for your work! 

Section 5, Conclusions 

better revisit many of them are from the discussion section; I suggest revisit it focusing on safety, productivity, and cost.

Author Response

Comments and Suggestions for Authors

Line 176 - give reference to your statement about sensitivity analysis! 

Line 183 - Section 2.2.2; how do you perform the sensitivity analysis, we want to know some references! I think it is inbuilt with the RS2D model, if so please mentioned it! and also discussed what the sensitivity analysis tells you!

Section 3, Results:

Line 203- earlier you mentioned that you are performing probabilistic analysis but how does the deterministic come here? Please correct it here or earlier or both with the new statement! Ok, you did both - hybrid!!! Li’s point-estimate is a probability approach.

Section 3.1.1, Line 227-228- do you have any reason or have any reference for taking 1.5 FOS as threshold slope? provide ref. 

Line 232 - check "slope angle beyond 500" is correct!

Figure 5, Line 345-Give sub-script for each figure and describe what the fig. is telling.

Figure 7, Line 262 - same as above in Fig. 5.

Figure 9, 10 - same as above Figs.

Section 3.2, Line need ref. for the Z-table;

Section 3.3, I think you need to mention the value of your mineral extracted and show the differences in the table with corresponding overburden or spoils to be removed! The graph is fine but this is not able to explain what you are saying!

Section 4, Discussion - I don't see any references in this section; go ahead and find some work relevant to this work and made a comprehensive discussion and how others are dealing with this type of issues!

Line 304, this line is repeated earlier in the results section, remove it or needed to restructure;

Line 310, Give the reason why the probability of failure due to rock mass movement was not estimated? I think this is important for your work! 

Section 5, Conclusions 

better revisit many of them are from the discussion section; I suggest revisit it focusing on safety, productivity, and cost.

[54, 55]

Yes, it is built-in function with the RS2D software. Conducting mesh sensitivity is highly recommended in finite elements to avoid the negative effects of very long/thin elements on analysis results. The aspect ratio of such thin elements is very high. The mesh quality option is a built-in feature in RS2D that allows users to identify and fix finite element mesh problems. Non-convergence of solution, terminated calculation, alerting or misconfigurations during computation, and extraordinary analysis results are examples of such issues [54, 55].

RS2 can automatically locate and highlight elements in a mesh that are deemed to be of "poor" quality based on user-definable criteria to assist the user in determining the "quality" of a finite element mesh. This is accomplished by selecting the Show Mesh Quality and Define Mesh Quality options from the Mesh Quality sub-menu.

Ok.

Please see reference [57]

Corrected: 500

Done

Done

Done

[56, 58]

Table 4 is added.

Some citations are included.

Removed

The probability of failure due to rock mass movement, on the other hand, will not be estimated because establishing a threshold is difficult. The use of rock mass deformation would be beneficial if multi-point hole extensometers were installed to monitor and read the rock mass deformation. As a result, it can be used to validate the readings with numerical analysis.

Done

Reviewer 2 Report

The proposed methodology for estimating the overall slope angle is an interesting one, but there are some considerations to be done.

The influence of overall slope angle on stripping ratio and implicit on profitability of open pit mining activity is a well-established fact. This influence is greater in the case of ore bodies with a great vertical development and decreases in the case of quasi-horizontal stratiform deposits.

For this reason, is important to apply the proposed methodology on a real open pit. In this case you could compare the actual overall slope angle resulting from deterministic analysis with overall slope angle from probabilistic analysis and influence on stripping ratio. For a real open pit, the profitability comparison will be more soundness.

In a real case analysis is important to take into account all influence factors besides the rock mass properties (e.g., seismicity, groundwater level, distributed load)

Regarding the idea to use backfill on open pit benches seems to be not so good. This supposes an additional distributed load with negative influence on factor of safety.

Author Response

Comments and Suggestions for Authors

The proposed methodology for estimating the overall slope angle is an interesting one, but there are some considerations to be done.

The influence of overall slope angle on stripping ratio and implicit on profitability of open pit mining activity is a well-established fact. This influence is greater in the case of ore bodies with a great vertical development and decreases in the case of quasi-horizontal stratiform deposits.

For this reason, is important to apply the proposed methodology on a real open pit. In this case you could compare the actual overall slope angle resulting from deterministic analysis with overall slope angle from probabilistic analysis and influence on stripping ratio. For a real open pit, the profitability comparison will be more soundness.

In a real case analysis is important to take into account all influence factors besides the rock mass properties (e.g., seismicity, groundwater level, distributed load)

Regarding the idea to use backfill on open pit benches seems to be not so good. This supposes an additional distributed load with negative influence on factor of safety.

We both agree. Your suggestions are greatly appreciated and has been considered. Please keep in mind that neither the deterministic nor the probabilistic analyses provided the overall slope angle. Slope geometry (e.g. angle, bench height and width) is the design factor you select.  A real open pit case study was subjected to deterministic and probabilistic analyses, which were then compared. Net profit is calculated for this open pit case study, with the exception of the cost of removing one ton of overburden and recovering one ton of ore (as no info is released).

Groundwater and seismicity have been examined.

We both agree. The suggestion has been removed.

Reviewer 3 Report

The paper deals with the problem of optimal slope angle of open pit mine. Its instability can be very serious problem from economic, safety and environmental point of view. Therefore, the probabilistic analysis performed by the Authors seems to be an important issue.

The paper consists of six parts. The first one introduces the subject and short references review is provided here. Authors divided the introduction into subparts discussing probabilistic methods separately. In my opinion chapter cannot have only one subchapter so either second one can be created or all text should be treated as one without its division.

Second part shows methods of slope stability analysis. Authors introduce Limit Equilibrium Methods (LEMs), analytical and finite-elements (FEMs) finite differences (FDMs) and discrete element methods (DEMs). Next, Shear Strength Reduction Technique (SSRT) is introduced. This approach takes safety as the main criterion into account. Authors used numerical modelling by means RocScience RS2D software. Model geometry as well boundary conditions for modeling were provided. Additionally, sensitivity analysis was performed to determine the optimal mesh size.

In third part Authors presented the results for the experiments conducted in accordance to the methodology presented in part two. Here, Authors presented the comparison between deterministic and probabilistic analyses. This was done for the factor of safety, maximum shear strain, incremental horizontal and vertical displacements. Then, the estimation of probability of failure was done for overall slope angle. Finally, the analysis of productivity and mining costs was done.

Fourth part is dedicated to discussion, where the main observations and notifications were presented basing on the results presented in previous chapter. This was the basis for the final conclusions presented in part five. The final part shows the main directions for future works.

Generally, the paper is well constructed. The introduction part requires a little bit of changes, but the further part is well presented. The results give wide spectrum of analysis and the conclusions are strongly supported with the presented graphs and numbers. The paper can be published in Applied Sciences after minor revision.

Author Response

Comments and Suggestions for Authors

The paper deals with the problem of optimal slope angle of open pit mine. Its instability can be very serious problem from economic, safety and environmental point of view. Therefore, the probabilistic analysis performed by the Authors seems to be an important issue.

The paper consists of six parts. The first one introduces the subject and short references review is provided here. Authors divided the introduction into subparts discussing probabilistic methods separately. In my opinion chapter cannot have only one subchapter so either second one can be created or all text should be treated as one without its division.

Second part shows methods of slope stability analysis. Authors introduce Limit Equilibrium Methods (LEMs), analytical and finite-elements (FEMs) finite differences (FDMs) and discrete element methods (DEMs). Next, Shear Strength Reduction Technique (SSRT) is introduced. This approach takes safety as the main criterion into account. Authors used numerical modelling by means RocScience RS2D software. Model geometry as well boundary conditions for modeling were provided. Additionally, sensitivity analysis was performed to determine the optimal mesh size.

In third part Authors presented the results for the experiments conducted in accordance to the methodology presented in part two. Here, Authors presented the comparison between deterministic and probabilistic analyses. This was done for the factor of safety, maximum shear strain, incremental horizontal and vertical displacements. Then, the estimation of probability of failure was done for overall slope angle. Finally, the analysis of productivity and mining costs was done.

Fourth part is dedicated to discussion, where the main observations and notifications were presented basing on the results presented in previous chapter. This was the basis for the final conclusions presented in part five. The final part shows the main directions for future works.

Generally, the paper is well constructed. The introduction part requires a little bit of changes, but the further part is well presented. The results give wide spectrum of analysis and the conclusions are strongly supported with the presented graphs and numbers. The paper can be published in Applied Sciences after minor revision.

We both agree. Thank you for your advice. The entire text has been combined and treated as a single chapter, with no divisions.

Round 2

Reviewer 2 Report

Be consequent with measure units in table 1 and 5.

Regarding the net profit, it must also take into account the cost of extracting the ore in addition to the cost of excavating the overburden.

Author Response

Comments and Suggestions for Authors

Be consequent with measure units in table 1 and 5.

Regarding the net profit, it must also take into account the cost of extracting the ore in addition to the cost of excavating the overburden.

Done. Line 167 (Table 1).

Done. Lines 388-392:

It is important to note that the total operating cost includes both the cost of ore extraction (e.g., $320×106) and the cost of overburden removal (e.g., $23.5×106). As a result, the total operating cost is 343.5×106. I f the selling price of one ton of ore is $200, then the total selling price will be $6,4×109.  Consequently, the net profit equals total selling price minus total operating costs (e.g., $64×109 - $343.5×106 = $63.6565×109).
